# Analysis of social combinations of COVID-19 vaccination: Evidence from a conjoint analysis

**Hanako Ohmura**  *

School of Policy Studies, Kwansei Gakuin University, Sanda-shi, Hyougo-ken, Japan

* hanakohmura@kwansei.ac.jp

## Abstract

Using a conjoint analysis based on Japanese cases, this study attempts to identify a preferable social strategic combination of who are vaccinated, who are not, and who are waiting. Using two surveys that relied on quota sampling reflecting the Japanese demographic composition ($n$ = 1024 & $n$ = 2975), the results of the descriptive analysis show that the most preferred strategy at the individual level was wait-and-see, allowing for a risk assessment of side effects. Via conjoint analysis, I also found that participants who recalled blood relatives as their familiar entities tended to prefer a wait-and-see strategy for themselves and their blood relatives. The results of these analyses suggest that wait-and-see strategies for vaccination are preferred in Japan, making it difficult to achieve early herd immunity through vaccination.

**Data Availability Statement:** All data and R-codes files are available from the Harvard Dataverse at doi:10.7910/DVN/YU7CC1 (https://dataverse. harvard.edu/dataset.xhtml?persistentId=doi:10. 7910/DVN/YU7CC1).

## Introduction

With the spread of the coronavirus disease (COVID-19), its consequent deaths, and the aftereffects in several countries, the best hope for mass immunization is vaccine development and administration. However, in society, individual intent to receive vaccination remains low. Representative studies show that the percentage of individuals willing to be vaccinated is 53.6% (undecided: 14.4%, unwilling: 32%) in the United States [1] and 73.9% (undecided: 18.9%, unwilling: 7.2%) in European countries [2], which are the main suppliers or initiators of vaccinations. As reported by Yoda and Katsumata [3], among the 1,100 online survey participants in Japan, 65.7% were willing to be vaccinated, 22.0% were not sure, and 12.3% were unwilling. Further, differences in the vaccination intentions between men and women and among people with different education levels have been investigated: vaccination intention increases 1.13 times in men with a higher level of education, whereas it decreases about 0.83 times in women with the same level of education [4].

Several studies have been conducted based on surveys related to intention to be vaccinated. Studies examining vaccination intent in more detail aim to determine the vaccine properties that individuals prefer, immunization protocols in place, and willingness of individuals to be vaccinated according to these situations [5–7]. These studies adopted discrete choice experiments [6] or conjoint analyses [7] to identify the vaccine property preferences of individuals and to detect the requirements for increased vaccine uptake. In addition to research on these

**Funding:** This work was supported by Grant-in-Aid for Young Scientists 19K13615 and Grant-in-Aid for Scientific Research (B) 21H00682.

**Competing interests:** The authors have declared that no competing interests exist.

vaccines as "one product," using these methods, it may be necessary to consider the influence of vaccination status in the society on the intention of individuals to be vaccinated. One problem that previous studies do not address is the need to examine the status of strategic interactions, namely, preferences of who should be inoculated and who should not, with exception of the research on healthcare [8].

While most individuals value scientific evidence and find it desirable that they are vaccinated altruistically, as well as the entire community, the most desirable combination of vaccinations among themselves, familiar individuals, and the society overall can occur in various ways [9]. In addition, some individuals may think strategically, wishing to "receive the vaccine after confirming the mid-/long-term adverse reactions," rather than a binary choice of simply wishing to receive it or not. Considering this nuanced alternative can also capture certain egoistic (individually rational) attitudes, such as "society as a whole should be vaccinated early, and I should be vaccinated after confirming the side effects and the effect on the variant." A strategic conflict between altruism and egoism is likely to arise in a society that provides vaccines, requiring a framework to explain the consequences of vaccination.

Similar to previous studies, this study uses a choice-based conjoint analysis for vaccination, to identify the types of strategic vaccine combinations most desirable in the Japanese society. The expectation is that the Japanese case could provide meaningful insights. Japan is a latecomer to the vaccine market, only starting to vaccinate healthcare professionals on February 18, 2021. Thus, vaccination has begun with less uncertainty regarding side effects than in other countries. As in the aforementioned [3, 4], the country has a high proportion of individuals adopting a "wait-and-see policy" as the US and European countries started vaccinations earlier. Japan's case has important implications to ascertain the social impact of the vaccination program, as many Japanese individuals are aware regarding the vaccine's side effects, with a low level of uncertainty about the vaccine.

This study can shed light on how people's egoistic or altruistic intentions may be behind the lack of sufficiently high vaccination intention in areas where COVID-19 is not widespread. Nevertheless, the implication is that, to achieve higher vaccination coverage, which is more medically desirable, adequately spreading information about the safety of vaccines to each generation is crucial.

## Empirical strategy

### Study participants and period

This study aims to identify vaccine combinations by introducing conjoint analysis into an online survey. Two online survey experiments were conducted, guiding survey panels from Yahoo Crowd Sourcing, Inc. (YCS) (March 14–16, 2021) and Lucid Holdings, LLC. (March 26–28, 2021). Lucid included 1,024 participants and 27 questions, while YCS included 2,975 participants and 43 questions. The selection of participants was based on quotas allocated according to demographic composition. As the further information, Supplemental Materials contain descriptive statistics on respondent variables and demographic composition compared to the census.

In addition, considering the inoculation schedule, I regard this survey period appropriate for verifying Japan's vaccination status. The vaccination of healthcare workers began on February 18, 2021, and I conducted my studies after about one month. On March 2, it was reported that a woman in her 60s died of a subarachnoid hemorrhage after vaccination. On March 12 (2 days before the YCS survey and 10 days before the Lucid survey), at a meeting of the vaccine study group held by the Ministry of Health, Labour and Welfare, 17 cases of adverse reactions caused by anaphylactic shock were reported. Considering these series of

events, I conducted my survey at the approximate time when Japanese citizens had already decided upon vaccination, that is, when more information about vaccination *per se* and its side effects was available.

## Ethics

This study was undertaken and approved by the Kwansei Gakuin University Committee for Regulations for Behavioral Research with Human Participants (27th February, 2021). In accordance with the Committee's recommendations, the participants were informed at the beginning of the survey that they may refuse to be presented with sensitive information about COVID-19 and may leave the investigation at any time. They were also informed at the debriefing that, if they felt uncomfortable with the information they received, they could opt not to send in their responses. The compensation for the survey was set to 20 Yahoo points for YCS and 3.8 USD for Lucid.

For details of the consent by the participants and their debriefing, all the survey questionnaires are available in the Supplementary Materials. The survey questionnaire was presented in Japanese to the Japanese participants, but an English translation is available in the Supplementary Materials.

## Design of the conjoint analysis

The contrivance in this conjoint analysis is as follows. First, to prevent an attribute of the conjoint analysis from becoming quite complicated, the pattern of attributes was constituted into three main bodies of "myself," "familial presence," and "society as a whole." The participants were asked to select one of the following for "familial presence," as a *reference group*: a family member who is older than the participant (e.g., parents), a family member who is younger than the participant (e.g., children), friend, colleague, neighbor, spouse, and significant other/ partner. For the criterion for distinguishing familiar individuals, I refer to the reference group setup in a study of relative income with the hypothetical choice experiment by [10, 11]. The participants were instructed to face the conjoint, while recalling a selected alternative as a familiar presence.

Second, the level for each attribute should not be a binary choice between vaccination and non-vaccination, and a third option should be incorporated, namely, "Do not vaccinate now, vaccinate later." This alternative allows us to capture certain egoistic attitudes such as a watcher, "society as a whole should be vaccinated early, and we should be vaccinated after confirming the occurrence of side effects." If the effect of *a watcher* is greater than that of the simple desire for "vaccinate," the acquisition of herd immunity through the vaccine is not necessarily an optimistic scenario.

According to the above conjoint analysis settings, I set my design of conjoint analysis as in Table 1 and Fig 1. Fig 2 shows an example of the conjoint screen.

I performed conjoint analysis using the Conjoint Survey Design Tool and by introducing the conjoint program into Qualtrics, according to the procedure by [12]. By using the method in [12], the effect of the concerned attribute X can be measured under all other attributes as

**Table 1. Proportion of willingness to be vaccinated.**

| Attributes | Values | | |
|---|---|---|---|
| Myself | Vaccinate | Do not vaccinate now, vaccinate later | Not vaccinate |
| Familiar presence | Vaccinate | Do not vaccinate now, vaccinate later | Not vaccinate |
| Society as a whole | Vaccinate | Do not vaccinate now, vaccinate later | Not vaccinate |

Q. Which of the following individuals are the most familiar to you? Please select one option and memorize it.

• Family members living with you who are older than you (e.g., parents)
• Family members living with you who are younger than you (e.g., children)
• Colleagues at work
• Neighbors
• Lover/Partner
• Spouse
• Friends

Explanations for conjoints

In the next question, you will be asked to select one of the two situations that you think is preferable regarding the COVID-19 vaccination. In this case, please assume and recall the option you selected in the previous question as the most familiar. There are five questions in total, so please make sure to answer all of them.

Conjoint experiments with 5 tasks

**Fig 1. Conjoint experiment flow.**

the average marginal component effect (AMCE), even if the effect of the attribute of interest is heterogeneous, with regard to the distribution of other attributes. Considering the examples in this study, [12] is able to measure the effect of the "myself" intention to vaccinate, based on the overall effects across other attributes: society in general and familiar entities.

## Results

Table 2 provides the simple descriptive statistics on the intention to be vaccinated. According to the YCS results, the most common response was "Do not vaccinate now, vaccinate later," followed by "vaccinate" and finally "not vaccinate." Conversely, Lucid's result implies that people prefer "vaccinate" compared to the wait-and-see strategy. However, many people in Japan still use the wait-and-see strategy. I may have obtained such results because I conducted the survey immediately after the reporting of specific information on adverse reactions, and it is likely that more citizens in Japan prefer the wait-and-see strategy compared to those in other countries. A major difference from the results of [3] is expected in the reporting of adverse reactions and the stabilization of infections during the investigation period.

Tables 3 and 4 show the results of the cross-tabulations of vaccination intentions by individual attributes. When examined by age, it is clear that the wait-and-see strategy is the most

Here, there are two options offered.

Which of the following two situations do you prefer?

Please select Choice 1 or Choice 2.

| | Choice 1 | Choice 2 |
|---|---|---|
| Myself | Do not vaccinate now, vaccinate later | Not vaccinate |
| Familiar person | Vaccinate | Do not vaccinate now, vaccinate later |
| Society as a whole | Vaccinate | Not vaccinate |

Choice 1

Choice 2

I don't know

I don't want to answer

**Fig 2. Display example of conjoint analysis.** *Note*: Five tasks were randomly displayed to participants.

common strategy for those in their 40s; however, it is also apparent that the vaccination intention is far greater than the intention to use the wait-and-see strategy, among those older than 50. This result is in line with the literature, which shows that vaccination intention increases with age [4]. Regarding sex, men clearly have higher vaccination willingness, and women are more likely to be inclined toward the wait-and-see strategy. In terms of educational level, the wait-and-see strategy was more prevalent among those with a four-year university degree or higher. This suggests that an increase in education level may strengthen the cautious attitude toward the wait-and-see strategy, rather than increasing the vaccination intention. Further, as support of a previous analysis [4], the results of the cross-tabulation between sex and education level, showed that the intention to be vaccinated was higher among men with a university

**Table 2. Proportion of willingness/wait-and-see/unwillingness to be vaccinated.**

| | Vaccination | Wait-and-see | No vaccination |
|---|---|---|---|
| YCS | 0.418 | 0.449 | 0.133 |
| Lucid | 0.462 | 0.401 | 0.137 |

*Note*: Each number refers to a percentage value (%).

**Table 3. Proportion of willingness/wait-and-see/unwillingness to be vaccinated by demographic composition: YCS.**

| | | Vaccination | Wait-and-see | | No vaccination | | |
|---|---|---|---|---|---|---|---|
| Age | | | | | | | |
| | 20–29 | 0.016 | 0.033 | | 0.012 | | |
| | 30–39 | 0.067 | 0.094 | | 0.033 | | |
| | 40–49 | 0.147 | 0.173 | | 0.047 | | |
| | 50–59 | 0.117 | 0.108 | | 0.031 | | |
| | 60–69 | 0.062 | 0.034 | | 0.007 | | |
| | 70–79 | 0.010 | 0.008 | | 0.001 | | |
| Sex | | | | | | | |
| | Female | 0.136 | 0.180 | | 0.057 | | |
| | Male | 0.283 | 0.268 | | 0.075 | | |
| Education | | | | | | | |
| | Graduated 4-year institution | 0.254 | 0.261 | | 0.063 | | |
| | Did not graduate | 0.165 | 0.189 | | 0.069 | | |
| Sex× | | Female | | | Male | | |
| Education | | Vaccination | Wait | No | Vaccination | Wait | No |
| | Graduated 4-year institution | 0.069 | 0.087 | 0.022 | 0.185 | 0.174 | 0.040 |
| | Did not graduate | 0.067 | 0.094 | 0.033 | 0.098 | 0.095 | 0.036 |

*Note*: Each number refers to a percentage value (%).

degree or higher, than the wait-and-see attitude. Conversely, the wait-and-see attitude was evident among women with a college degree. These results are common to both the YCS and Lucid results, with a more pronounced trend observed in the Lucid findings.

The majority of the respondents who answered "vaccinate later" or "no vaccinate" declared that they wanted to check the side effects of the vaccine. As predicted, the percentage exceeded

**Table 4. Proportion of willingness/wait-and-see/unwillingness to be vaccinated by demographic composition: Lucid.**

| | | Vaccination | Wait-and-see | | No vaccination | | |
|---|---|---|---|---|---|---|---|
| Age | | | | | | | |
| | 20–29 | 0.054 | 0.080 | | 0.030 | | |
| | 30–39 | 0.074 | 0.112 | | 0.019 | | |
| | 40–49 | 0.086 | 0.104 | | 0.031 | | |
| | 50–59 | 0.100 | 0.083 | | 0.025 | | |
| | 60–69 | 0.099 | 0.032 | | 0.011 | | |
| | 70–79 | 0.047 | 0.011 | | 0.002 | | |
| Sex | | | | | | | |
| | Female | 0.174 | 0.238 | | 0.073 | | |
| | Male | 0.282 | 0.186 | | 0.047 | | |
| Education | | | | | | | |
| | Graduated 4-year institution | 0.279 | 0.224 | | 0.0655 | | |
| | Did not graduate | 0.176 | 0.200 | | 0.066 | | |
| Sex× | | Female | | | Male | | |
| Education | | Vaccination | Wait | No | Vaccination | Wait | No |
| | Graduated 4-year institution | 0.084 | 0.116 | 0.030 | 0.196 | 0.109 | 0.023 |
| | Did not graduate | 0.090 | 0.123 | 0.042 | 0.086 | 0.078 | 0.023 |

*Note*: Each number refers to a percentage value (%).

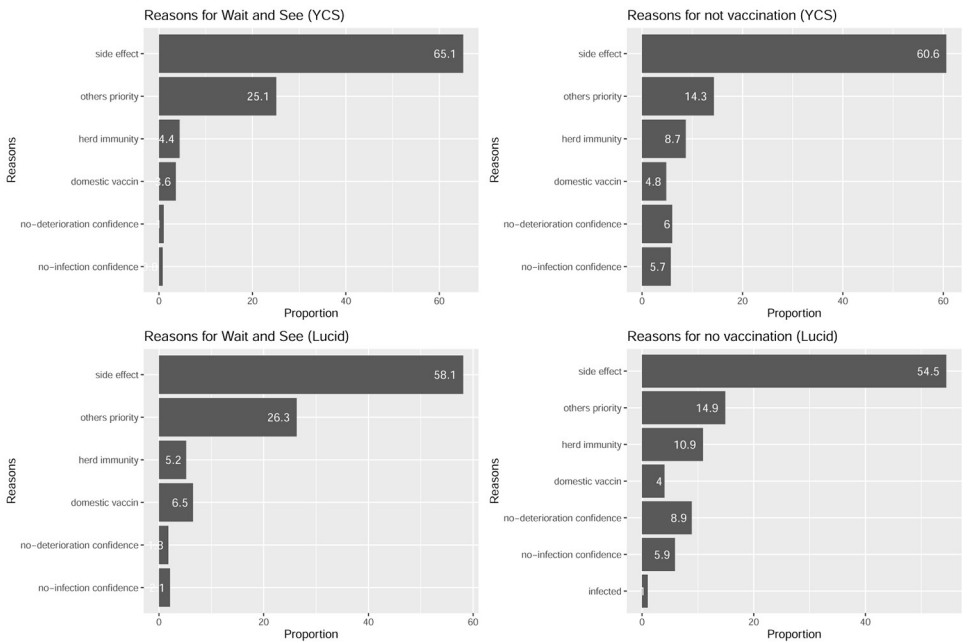

**Fig 3. Reasons for not getting vaccinated and watching.** *Note*: Abbreviations: side effects = I want to see if there are any adverse reactions to the vaccine. Others' priority = The product of the desired vaccine manufacturer is not available in Japan. Herd immunity = If others are inoculated first and herd immunity is established, there is no need to inoculate myself as soon as possible. No-deterioration confidence = Even if I am infected, it is unlikely that I will become seriously ill. No-infection confidence = I will not be infected. Infected = I have already been infected with COVID-19.

50% in all surveys (Fig 3). Another is an altruistic reason: "there are people who should get vaccinated before me." Finally, the third most common reason was "once herd immunity is established, it is not necessary to inoculate yourself," which was marked as a more strategic and egoistic intent.

I then examined the results of a pooled conjoint analysis of all participants. Fig 4 supports the descriptive analysis, revealing that "wait-and-see" is the preferred strategy for self and

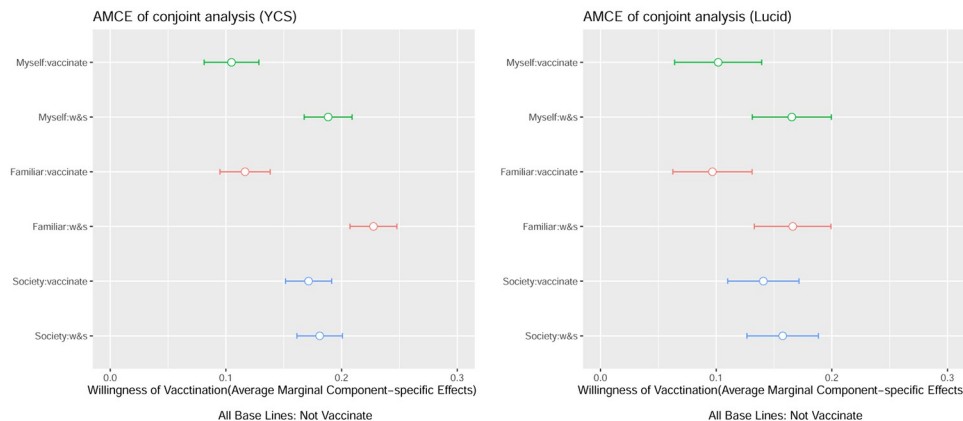

**Fig 4. Average marginal component effect (AMCE) results.** *Note*: This plot shows the estimates of the effects of the randomly assigned vaccination attribute values on the probability of being preferred by Japanese participants. The estimates are based on the benchmark OLS model with clustered standard errors, and the bars represent the 99% confidence intervals. The points denote the attribute value, which is the reference category for each attribute.

personal existence (baseline: not vaccinate). It is difficult to find significant differences between the overall results of Lucid's panel in the AMCEs because of the small sample size. However, there are significant differences among AMCEs, namely, vaccination, watching, and non-vaccination, especially in the case of familiar entities. Furthermore, for the society in general, the results show no noticeable difference in the effects of vaccination among watchers. However, for the society in general, the results show no difference between static and active inoculation. Both results for familiar entities and the society suggest that the *altruism* of "withholding vaccination for the time being" seems to work, not for the society in general, but only for other people who are particularly close to each other. The results of this study show no support for the altruism in a pure sense, which is observed in the intention to contribute to herd immunity by actively inoculating oneself. More importantly, the wait-and-see strategy is clearly preferred for one's self and those who have close relations, whereas the watcher strategy for the society in general is not preferred.

Next, I examined the conjoint results relating to familiar entities. As shown in Fig 5, the most common choices, in descending order, were as follows: spouse and then family members living together, who were older and younger than the participants. This order was the same in both studies. Clearly, close family was the most frequent choice. In this context, Fig 6 shows the results of the conjoint, sorted according to who is selected as a familiar entity, and its baseline is set as 'not to be inoculated'. Strikingly, those who select younger family members are less likely to prefer vaccinations for familiar entities and are more likely to hope that younger members are watchers. Furthermore, even the group that selected elderly people who are living family members, as the highest risk group at the time of infection, did not intend to receive vaccination, among those who were close to them (especially, in Lucid's case), and the wait-and-see strategy played a central role.

## Conclusions

With fewer COVID-19 deaths in Japan than in other countries, the intention to receive vaccination is low, with individuals adopting the "wait-and-see" strategy. Results of a conjoint analysis that delved further into intentions, showed a strong preference for the wait-and-see strategy, not only for the participants themselves but also for those with whom they were familiar. This suggests that altruism does not lead to herd immunity through active voluntary vaccination; instead, a reduction in the side effects of vaccination leads to increased willingness to inoculate, the objective being to protect oneself or one's friends and families. In addition, this study reveals that the preference of a familiar person to adopt the wait-and-see strategy was stronger in those who recall blood relations as familiar entities.

Clearly, in the case of Japan, the altruism surrounding vaccines did not manifest as a psychological mechanism for wanting the vaccination to be given as soon as possible to close relatives, but rather as a mechanism to wait until one is fully aware of the side effects. Beyond simple altruism, egoism for the safety of close relatives has been also embossed from this research. In countries where vaccination is not widely spread, the longer the observation period for side effects, the more likely it is that the strategy of observing the consequences and waiting until the safety from side effects is sufficiently assured, will be adopted. Hence, my analysis indicates that the most preferred social combinations of vaccination are as follows: society in general, may or may not be vaccinated early, but my loved ones and I can wait long enough to be vaccinated.

Whatever the reason for the wait-and-see behavior and non-vaccination, these results suggest that it would be difficult to establish herd immunity by increasing the number of vaccinated individuals at an early stage. To suppress opportunistic strategies and cultivate

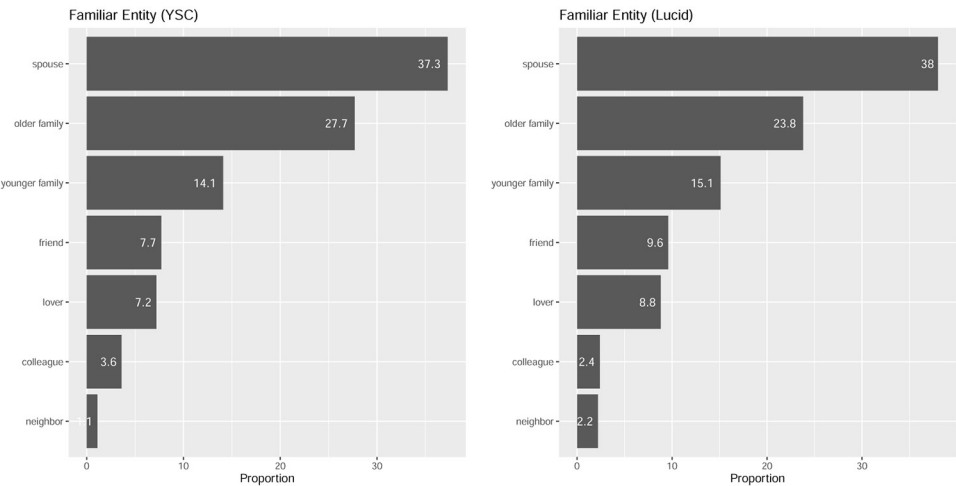

**Fig 5. Selection of familiar entities.**

intentions for mass vaccination, the public should be fully informed that the risk of infection is higher than the risk of side effects. Furthermore, considering the result that people are more cautious about allowing their relatives who are from a different generation to receive vaccination, proving the safety of vaccines for each generation and publicizing its safety, is an efficient way to enhance the intention to be vaccinated. However, even with such countermeasures in place, empirical results pessimistically predict that vaccination rates would remain at a low level among Japanese citizens.

The first limitation of this study, however, is that the results of this analysis are limited to Japanese participants, and there is a possibility that other factors may function as the main motivation behind the adoption of the wait-and-see and non-inoculation strategies in other countries (e.g., some campaign discourse on vaccine toxicity). For the second limitation, as

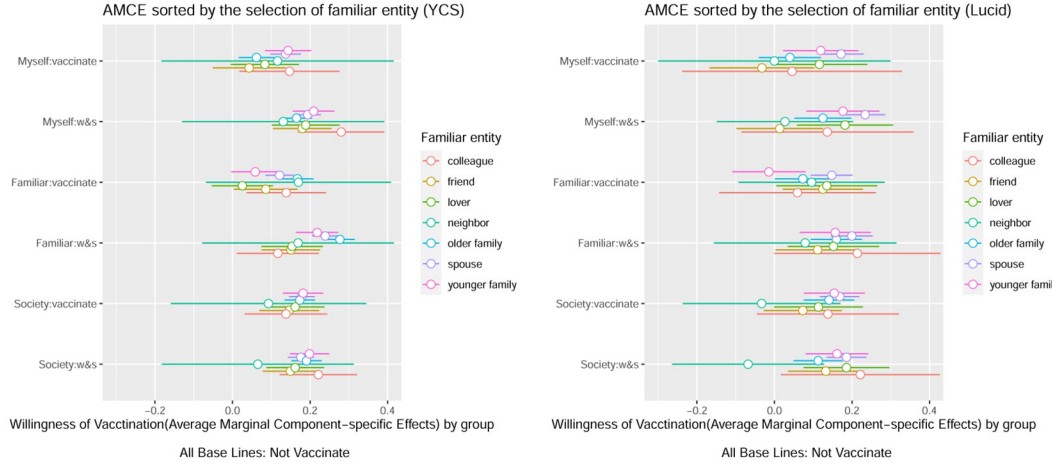

**Fig 6. Average marginal component effect (AMCE) results by group.** *Note*: This plot shows the estimates of the effects of the randomly assigned vaccination attribute values on the probability of being preferred by Japanese participants. Coefficient plots are sorted by groups, as per familial presence recalled by participants. The estimates are based on the benchmark OLS model with clustered standard errors, and the bars represent 99% confidence intervals. The points denote the attribute value, which is the reference category for each attribute.

this study did not handle panel data obtained from several waves of surveys, the design may not be sufficiently controlled for covariates (unit-specific effects). To address such challenges, this paper relies on the results of analyses using multiple surveys at roughly the same time point. However, the limitation is that it still fails to capture the changes that occur over time within each unit.

As of October 20, 2021, fortunately, against the implication of this study, 7 months after the survey was conducted in March 2021, the percentage of those who have received at least one shot of vaccination and the percentage of those who have received two shots in Japan were 76% and 67.8%, respectively (Source: *Nikkei Shimbun* HP about Vaccination Status URL: https://vdata.nikkei.com/newsgraphics/coronavirus-japan-vaccine-status/), far exceeding the levels predicted by the results of the analysis in this study. Research suggesting that vaccination reduces the number of effective reproductions ($R(t)$) has, is ongoing [13]. Compared to the results of the survey on vaccination intention in the U.S., as of October 2020 (i.e., 53.6% [1]), and the vaccination rate as of October 2021 (i.e., 57.5%), the prior vaccination intention and actual vaccination rates are close (Source: *New York Times* URL: https://www.nytimes.com/interactive/2020/us/covid-19-vaccine-doses.html). In the EU, the percentage of the completion of partial vaccination, as of September 2021, was 60s in Western Europe and around 20% in Eastern Europe, compared to prior vaccination intention of over 70% (Source: European Centre for Disease Prevention and Control URL:https://www.ecdc.europa.eu/en/publications-data/data-covid-19-vaccination-eu-eea)Compared to countries where these prior vaccination intentions and actual vaccination rates are consistent, and where actual vaccination rates remain low when compared to prior vaccination intentions, the question of why the Japanese society was able to overcome its initial low vaccination intentions and reach a high vaccination completion rate remains; together with the question of why this enables Japan to act to lower $R(t)$, these are issues to be unraveled by future research and comprise the third limitation of this study. The hypotheses for these questions include (1) a vigorous public outreach program with medical experts, (2) a vaccination system that was quickly promoted by both local public administrative bodies and corporate companies, and (3) the contribution of social peer pressure as an opportunity to transform individuals' vaccination intentions. Thus, additional surveys are needed to test these hypotheses in future research.

## Supporting information

**S1 Table. Supplemental R-code for descriptive statistics and demographic composition.**
(R)

**S1 Fig. Example of a display screen of conjoint analysis in the original Japanese language.**
(PDF)

**S2 Fig. Supplemental data for Fig 3.**
(CSV)

**S3 Fig. Supplemental data for Fig 3.**
(CSV)

**S4 Fig. Supplemental data for Fig 3.**
(PDF)

**S5 Fig. Supplemental R-code for Fig 3.**
(R)

**S6 Fig. Supplemental data for Fig 4.**
(R)

**S7 Fig. Supplemental data for Fig 4.**
(R)

**S8 Fig. Supplementary materials for Fig 4.**
(PDF)

**S9 Fig. Supplemental R-code for Fig 5.**
(R)

**S10 Fig. Supplemental data for Fig 6.**
(R)

**S11 Fig. Supplemental data for Fig 6.**
(R)

**S1 Questionnaire. The survey questionnaire can be accessed here.**
(PDF)

## Acknowledgments

The author thanks Takeshi Iida, Tetsuya Matsubayashi, and Go Murakami for the discussions that helped for the motivation this study.

## Author Contributions

**Conceptualization:** Hanako Ohmura.

**Data curation:** Hanako Ohmura.

**Formal analysis:** Hanako Ohmura.

**Funding acquisition:** Hanako Ohmura.

**Investigation:** Hanako Ohmura.

**Methodology:** Hanako Ohmura.

**Project administration:** Hanako Ohmura.

**Resources:** Hanako Ohmura.

**Software:** Hanako Ohmura.

**Validation:** Hanako Ohmura.

**Visualization:** Hanako Ohmura.

**Writing – original draft:** Hanako Ohmura.

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
