## [Decision Letter · Decision Letter 0]

19 Oct 2021

PONE-D-21-21028

Analysis of Social Combinations of Coronavirus Vaccination: Evidence from a Conjoint Analysis

PLOS ONE

Dear Dr. Ohmura,

Thank you for submitting your manuscript to PLOS ONE. After careful consideration, we feel that it has merit but does not fully meet PLOS ONE’s publication criteria as it currently stands. Therefore, we invite you to submit a revised version of the manuscript that addresses the points raised during the review process.

We look forward to receiving your revised manuscript.

Kind regards,

Prof. Anat Gesser-Edelsburg, Ph.D.

Academic Editor

PLOS ONE

“Corresponding author: Professor, School of Policy Studies, Kwansei Gakuin University. 2-1, Gakuen, Sanda-shi,

Hyougo-ken, Japan, 6691337. hanakohmura@kwansei.ac.jp. This work was supported by Grant-in-Aid for Young

Scientists, Japan Society for Promotion of Science 19K13615”

“This work was supported by Grant-in-Aid for Young Scientists 19K13615.”

Additional Editor Comments (if provided):

Reviewers' comments:

Reviewer's Responses to Questions

**Comments to the Author**

1. Is the manuscript technically sound, and do the data support the conclusions?

Reviewer #1: Yes

Reviewer #2: Partly

2. Has the statistical analysis been performed appropriately and rigorously? 

Reviewer #1: Yes

Reviewer #2: Yes

3. Have the authors made all data underlying the findings in their manuscript fully available?

Reviewer #1: Yes

Reviewer #2: Yes

4. Is the manuscript presented in an intelligible fashion and written in standard English?

Reviewer #1: Yes

Reviewer #2: Yes

5. Review Comments to the Author

Reviewer #1: The article presents a very current and necessary topic. It is well written and has a great methodological quality. The only suggestion I have is:

- in the introduction, provide an international overview of the subject.

- improve the limitations of the study.

- the practical and/or clinical implications of the study need to be further emphasized. It should make clearer the importance of the study for society and the next steps for the future.

Reviewer #2: Manuscript Number: PONE-D-21-21028

Manuscript Title: Analysis of Social Combinations of Coronavirus Vaccination: Evidence from a Conjoint Analysis

October 17, 2021

Reviewer

Comments to the authors:

This study provided some important evidence on vaccination dynamics in Japan using a conjoint analysis to breakdown the dynamics into groups. This is very important to determine the attributes of the groups in this pandemic. The aim of this study is laudable and evidentially justified. The statistical analyses, results, and discussions were aligned, which helped in understanding and reviewing this paper.

Revise grammatical errors in the manuscript throughout.

I would recommend the following:

Abstract:

-Provide the sample size and indicate whether this is a nationally representative sample or not.

-Are these results statistically significant or based on descriptive statistics? Please include this information.

-What conclusion(s) could you draw from these findings? I would suggest including this information.

-Check your quotations. Some of the quotations have apostrophes in addition. The quotation marks are also different.

Introduction

-Paragraph one: What are the vaccination statistics in Japan? Provide the vaccination statistics in Japan to compare these information to that of other countries you mentioned, instead of continents.

-Paragraph two: Rewrite this “(for example McPhedran and Toombs, 2021; Motta, 2021; Kreps et al., 2020)” as “(McPhedran and Toombs, 2021; Motta, 2021; Kreps et al., 2020)”.

-Paragraph four: Add this paragraph to paragraph five as its starting paragraph.

-Paragraph six: State your study aim and objectives here. No need stating “section” etc., as original research papers are organized in this manner.

Methods

Study subjects and period: Did you collect sociodemographic data on the subjects? If yes, I would recommend that you provide these information for us to better understand the dynamics of vaccination by subgroups.

Ethics: Was ethical approval obtained? This was not clearly stated.

Design of conjoint analysis: Please revise the grammar at this section. You were using both past tense and present tense. For instance, you used a present tense in “Subjects are asked to select one of the following, . . .” but used a past sentence in “The subjects were instructed. . .”

-Revise this sentence “According to the above conjoint analysis settings, we set our design of conjoint as in Table 1 and Figure 1, and Figure 2shows an example of the conjoint screen” as “According to the above conjoint analysis settings, we set our design of conjoint as in Table 1 and Figure 1, and Figure 2 shows an example of the conjoint screen.”

Results

-Paragraph two: Tables 1 and 2 should be formatted to remove a lot of the borderlines.

-What could be some possible limitations to your study? It is important to acknowledge this in a research.

-Provide some potential empirical studies that could have explained vaccination dynamics or perceptions about vaccination. You could also identify studies that agree or disagree with your findings.

Conclusion

No comments

Thank you very much for your important work.

6. PLOS authors have the option to publish the peer review history of their article (what does this mean?). If published, this will include your full peer review and any attached files.

Reviewer #1: **Yes: **Mateus Dias Antunes

Reviewer #2: No

---

## [Author Response · Author response to Decision Letter 0]

16 Nov 2021

Reviewer #1: I would like to express my sincere gratitude for the favorable review of my manuscript and would appreciate it if you could review my responses to the three points as follows.

- in the introduction, provide an international overview of the subject.

In addition to the descriptions of the U.S. and EU for international comparisons in the Introduction section, I have added statistics on vaccination intentions for Japan, corresponding to the point raised by Reviewer 2. In the Conclusions section, I have included a description about a cross-comparison of vaccination dynamics in Western countries and Japan. These newly added descriptions are founded upon international comparisons in previous studies rather than simply presenting the actual situation based on statistics.

- improve the limitations of the study

As a limitation of this study, I first note in the Conclusion section that the findings of this study are limited to subjects in Japan. Also, as a limitation of this study and a suggestion for new research, I have also reported that the current vaccination dynamics in Japan are so advanced that they have reached a situation that overturns the results of this analysis. Here, I also mentioned that a puzzle for new research has arisen.

- the practical and/or clinical implications of the study need to be further emphasized. It should make clearer the importance of the study for society and the next steps for the future.

As practical and clinical implications of this study, at the end of the Introduction and in the Conclusions sections, I have added the need to prove and publicize the safety of vaccines for each generation. In particular, since this survey was conducted in early March 2021, some studies have begun to show that the vaccination rate in Japan has increased significantly since then, and that this has pushed down the effective reproduction number (R(t)). It is also hypothesized that what contributed to this increase in vaccination rates and subsequent decrease in R(t) was the government’s publicity on TV and public commercials by experts about the limited side effects of vaccines. Since a detailed examination of these hypotheses is an issue for the future, a paragraph on the dynamics of vaccination in Japan and its implications has been added at the end of the Conclusions section.

 

Reviewer #2

I am very grateful for the positive review and the many important suggestions for improvement. The following responses address each of these points, and I hope you will find them helpful.

-Revise grammatical errors in the manuscript throughout.

Thank you for the suggestion. I have hired an English-language editing company to proofread the entire paper.

Abstract:

-Provide the sample size and indicate whether this is a nationally representative sample or not.

I have included a description of the sample size for the two surveys and how they match up to national demography.

-Are these results statistically significant or based on descriptive statistics? Please include this information.

Since the empirical results consist of descriptive statistics and a conjoint analysis with tests of statistical significance, I have rewritten the description of the results to distinguish between these analyses.

-What conclusion(s) could you draw from these findings? I would suggest including this information.

-Check your quotations. Some of the quotations have apostrophes in addition. The quotation marks are also different.

I have added the conclusion (and implications) of this study in the Abstract section and also corrected the quotation marks.

Introduction:

-Paragraph one: What are the vaccination statistics in Japan? Provide the vaccination statistics in Japan to compare these information to that of other countries you mentioned, instead of continents.

In response to this comment, I have added statistics on prior vaccination intentions for Japan. Rather than simply adding statistics, I have included statistics from studies such as Yoda et al. (2021) and Ishimaru et al. (2021) with the intention of introducing related studies.

-Paragraph two: Rewrite this “(for example McPhedran and Toombs, 2021; Motta, 2021; Kreps et al., 2020)” as “(McPhedran and Toombs, 2021; Motta, 2021; Kreps et al., 2020)”.

All literature citations have been changed to a number formation. All references have been renumbered, and vague descriptions such as “for example” have been removed.

-Paragraph four: Add this paragraph to paragraph five as its starting paragraph.

I would like to thank you for your useful suggestions. I have re-written the paragraphs as you suggested.

-Paragraph six: State your study aim and objectives here. No need stating “section” etc., as original research papers are organized in this manner.

I also would like to thank you for your useful suggestions. As you pointed out, I have presented the purpose of this study and a brief conclusion in the last paragraph of the Introduction.

Methods:

Study subjects and period: Did you collect sociodemographic data on the subjects? If yes, I would recommend that you provide these information for us to better understand the dynamics of vaccination by subgroups.

In accordance with this helpful comment, I have presented the results on vaccination intention according to sociodemographic characteristics in Tables 3 and 4. By discriminating between vaccination intentions of subgroups, I have added the following explanations: (1) women are more likely to use the wait-and-see strategy than are mem, (2) this tendency is more conspicuous for women with a higher education, and (3) the results are consistent with the findings of previous studies such as Ishimaru (2021). 

Ethics: Was ethical approval obtained? This was not clearly stated.

Design of conjoint analysis: Please revise the grammar at this section. You were using both past tense and present tense. For instance, you used a present tense in “Subjects are asked to select one of the following, . . .” but used a past sentence in “The subjects were instructed. . .”

-Revise this sentence “According to the above conjoint analysis settings, we set our design of conjoint as in Table 1 and Figure 1, and Figure 2shows an example of the conjoint screen” as “According to the above conjoint analysis settings, we set our design of conjoint as in Table 1 and Figure 1, and Figure 2 shows an example of the conjoint screen.”

Thank you for pointing out the grammatical errors. I have corrected these errors. In addition, ethical approval had been obtained. I have added those details to the paper.

Results

-Paragraph two: Tables 1 and 2 should be formatted to remove a lot of the borderlines.

As you suggested, I tried to see if I could remove the lines, but I think this is the limit of what I can do to maintain readability. I would appreciate it if you would consider it.

-What could be some possible limitations to your study? It is important to acknowledge this in a research.

I apologize for the inadequate description of the limitations of the study and thank you for your remarks. As the other reviewer also pointed out, we did not fully address the limitations of the study: one is that the findings are limited to Japan; so even if there is a large number of quiescent or non-vaccinated people in other countries, this may not be due solely to avoidance of side effects in blood relatives. In other countries, even if there are a large number of people who are watchers or do not want to be vaccinated, it may not be due only to avoidance of side effects on blood relatives. This is the first limitation of this study. 

 Surveys for this study were conducted in March 2021, and it has been almost 8 months since then. This has led to the discrepancy between the prior vaccination intention rates and actual ones. A relatively high vaccination rate is being achieved in Japan, even beyond the expectations of this study. This discrepancy partially constitutes a limitation of this study, but I recognize that it can be a critical future issue to be addressed. Thus, I have emphasized on answering why lower prior vaccination intentions are being overcome and higher vaccination rates are being achieved in Japanese society has been set as a future topic of research. To answer this question, three hypotheses are presented as well in the end of the Conclusion section.

-Provide some potential empirical studies that could have explained vaccination dynamics or perceptions about vaccination. You could also identify studies that agree or disagree with your findings.

Thanks to your comment, I followed up on recent studies on vaccination dynamics and, in relation to the above point, to mention the discrepancy between vaccination dynamics and the predictions of this study. In the Conclusions section, I have mentioned the need to examine why prior vaccination intentions and actual vaccination diffusion tend to coincide in the U.S. and European countries, whereas they tend to diverge in Japan.

---

## [Decision Letter · Decision Letter 1]

24 Nov 2021

PONE-D-21-21028R1Analysis of Social Combinations of COVID-19 Vaccination: Evidence from a Conjoint AnalysisPLOS ONE

Dear Dr. Ohmura,

Thank you for submitting your manuscript to PLOS ONE. After careful consideration, we feel that it has merit but does not fully meet PLOS ONE’s publication criteria as it currently stands. Therefore, we invite you to submit a revised version of the manuscript that addresses the points raised during the review process.

We look forward to receiving your revised manuscript.

Kind regards,

Prof. Anat Gesser-Edelsburg, Ph.D.

Academic Editor

PLOS ONE

Journal Requirements:

Reviewers' comments:

Reviewer's Responses to Questions

**Comments to the Author**

1. If the authors have adequately addressed your comments raised in a previous round of review and you feel that this manuscript is now acceptable for publication, you may indicate that here to bypass the “Comments to the Author” section, enter your conflict of interest statement in the “Confidential to Editor” section, and submit your "Accept" recommendation.

Reviewer #2: All comments have been addressed

2. Is the manuscript technically sound, and do the data support the conclusions?

Reviewer #2: Yes

3. Has the statistical analysis been performed appropriately and rigorously? 

Reviewer #2: Yes

4. Have the authors made all data underlying the findings in their manuscript fully available?

Reviewer #2: Yes

5. Is the manuscript presented in an intelligible fashion and written in standard English?

Reviewer #2: Yes

6. Review Comments to the Author

Reviewer #2: Thank you for addressing most of my comments.

I have the following minor comments:

You were using "I" but at some points in the method section, you were using "our" and "us". Please, be consistent.

Methods

Design of conjoint analysis: This sentence may not be complete, “According to the above conjoint analysis settings, I set my design of conjoint as in Table 1 and Figure 1, and Figure shows an example of the conjoint screen.” I think ". . . Figure shows an example of the conjoint screen." should have a number as ". . . Figure 2 shows an example of the conjoint screen."

Results

Lines 137-138: ". . . as predicted, the percentage exceeded 50% in all surveys (Figure )." The figure is missing a number.

Conclusion

The information you provided as a limitation does not reflect a study limitation. Your limitation should focus on the measures, study design and sampling, analytical procedures, etc. For instance, this is a cross-sectional study, so you could not establish causality and temporality.

7. PLOS authors have the option to publish the peer review history of their article (what does this mean?). If published, this will include your full peer review and any attached files.

Reviewer #2: No

---

## [Author Response · Author response to Decision Letter 1]

30 Nov 2021

Reviewer #2 

I would like to thank you for your second detailed review of this paper and for your valuable suggestions. I have responded to each comment as follows and would appreciate your confirmation.

You were using "I" but at some points in the method section, you were using "our" and "us". Please, be consistent.

I have ensured that all instances of “us” and “our” were deleted in this newly revised version.

Methods

Design of conjoint analysis: This sentence may not be complete, “According to the above conjoint analysis settings, I set my design of conjoint as in Table 1 and Figure 1, and Figure shows an example of the conjoint screen.” I think ". . . Figure shows an example of the conjoint screen." should have a number as ". . . Figure 2 shows an example of the conjoint screen."

Thank you for pointing this out. The figure number was indeed missing, so I have added it accordingly.

Results

Lines 137-138: ". . . as predicted, the percentage exceeded 50% in all surveys (Figure )." The figure is missing a number.

I would like to thank you for your careful review of this as well. I have added the figure number accordingly. I have also embedded each percentage value in the bar graph in the figures. I hope that the readability of the figures has improved.

Conclusion

The information you provided as a limitation does not reflect a study limitation. Your limitation should focus on the measures, study design and sampling, analytical procedures, etc. For instance, this is a cross-sectional study, so you could not establish causality and temporality.

I would like to thank you for your kind remarks. One of the limitations of this study is that the data were not panel data consisting of several waves, the unit specific effects were not controlled, and there was a difficulty in handling covariates. I would appreciate it if you could confirm this (l.204–l.212).

---

## [Editor Report · Decision Letter 2]

2 Dec 2021

Analysis of Social Combinations of COVID-19 Vaccination: Evidence from a Conjoint Analysis

PONE-D-21-21028R2

Dear Dr. Ohmura,

We’re pleased to inform you that your manuscript has been judged scientifically suitable for publication and will be formally accepted for publication once it meets all outstanding technical requirements.

Kind regards,

Prof. Anat Gesser-Edelsburg, Ph.D.

Academic Editor

PLOS ONE
---

## [Editor Report · Acceptance letter]

9 Dec 2021

PONE-D-21-21028R2 

Analysis of Social Combinations of COVID-19 Vaccination: Evidence from a Conjoint Analysis 

Dear Dr. Ohmura:

I'm pleased to inform you that your manuscript has been deemed suitable for publication in PLOS ONE. Congratulations! Your manuscript is now with our production department. 

Kind regards, 

on behalf of

Prof. Anat Gesser-Edelsburg 

Academic Editor

PLOS ONE